# Diverse Neutrophil Functions in Cancer and Promising Neutrophil-Based Cancer Therapies

**DOI:** 10.3390/ijms232415827

**Published:** 2022-12-13

**Authors:** Khetam Sounbuli, Nadezhda Mironova, Ludmila Alekseeva

**Affiliations:** 1Institute of Chemical Biology and Fundamental Medicine SB RAS, Lavrentiev Ave., 8, Novosibirsk 630090, Russia; 2Faculty of Natural Sciences, Novosibirsk State University, Novosibirsk 630090, Russia

**Keywords:** neutrophil heterogeneity, tumor-associated neutrophils, tumor microenvironment, cancer therapy

## Abstract

Neutrophils represent the most abundant cell type of leukocytes in the human blood and have been considered a vital player in the innate immune system and the first line of defense against invading pathogens. Recently, several studies showed that neutrophils play an active role in the immune response during cancer development. They exhibited both pro-oncogenic and anti-tumor activities under the influence of various mediators in the tumor microenvironment. Neutrophils can be divided into several subpopulations, thus contradicting the traditional concept of neutrophils as a homogeneous population with a specific function in the innate immunity and opening new horizons for cancer therapy. Despite the promising achievements in this field, a full understanding of tumor–neutrophil interplay is currently lacking. In this review, we try to summarize the current view on neutrophil heterogeneity in cancer, discuss the different communication pathways between tumors and neutrophils, and focus on the implementation of these new findings to develop promising neutrophil-based cancer therapies.

## 1. Introduction

Neutrophils represent the most abundant cell type of leukocytes in human blood and the second most in mice [1]. Neutrophils are named for their ability to be stained with a mixture of alkaline and acidic dyes [2]. Mature neutrophils are differentiated from hematopoietic stem cells in the bone marrow in a process called granulopoiesis and are produced in high quantities, up to 10^11^ per day in healthy individuals [3]. They are the first line of defense against pathogens, which explains the high susceptibility of people with neutropenia to infections [4]. Neutrophils were always considered a homogeneous population with specific functions in innate immunity, most likely due to their short life span, which limited the ability to investigate their diverse activities or even expect them. The recent observations of neutrophil heterogeneity in the steady state [5], in different tissues [6], and in pathology [7,8] have dramatically altered the old paradigm of neutrophil homogeneity. The recent reconsideration of neutrophil biology was achieved thanks to advances in biotechnology, which enabled researchers to investigate cells at a single-cell resolution [9]. In cancer, neutrophil actions are diverse and heterogeneous. Neutrophil blood levels increase during cancer progression [3]. Neutrophilia is associated with poor prognosis in many cancer types [10]. In addition to quantitative changes, qualitative changes in neutrophils upon cancer were observed. These changes include alterations in neutrophil morphology and function. The observation of tumor-associated neutrophils (TANs) producing neutrophil extracellular traps (NETs) was a hint of the possible role of neutrophils in the tumor microenvironment [11]. NETs, first observed by Brinkmann et al. in 2004, are web-like structures of neutrophilic genetic material decorated with the proteins of granules [12]. Later, NETs were shown to be involved in cancer metastasis [13]. In addition to NETs, neutrophils, after recruitment to the tumor microenvironment, could gain an anti-tumor (N1) or a pro-tumor (N2) phenotype [14]. Neutrophil polarization seems to be a complicated process affected by several tumor-derived factors. Besides this classification, a high percent of neutrophils in the circulation of cancer patients were shown to have a lower density (low-density neutrophils, LDNs) [15] and to exhibit some features of immaturity and immunosuppressive function (granulocytic-myeloid-derived suppressor cells (g-MDSCs) [16]. The recently described interactions between neutrophils and tumors prompted the scientific community to develop neutrophil-based cancer therapies. Achievements in this field are very promising and have reached the generation of chimeric antigen receptor neutrophils (CAR-neutrophils) [17].

Here, we summarize the different neutrophil populations observed in cancer in recent studies, review the interactions between neutrophils and tumor cells in the tumor microenvironment, and focus on novel neutrophil-based cancer therapies.

## 2. Neutrophil Heterogeneity in Cancer: N1/N2, NDN/LDN, and g-MDSC

### 2.1. N1 vs. N2

The story of neutrophil heterogeneity in cancer started with Fridlender et al.’s study, suggesting for the first time the N1/N2 functional classification of TANs. The authors introduced a new classification of neutrophils, analogous to the M1/M2 macrophage classification: N1—neutrophils with pro-inflammatory properties and anti-tumor functions, and N2—neutrophils with anti-inflammatory and pro-tumor functions [14]. Various factors influence the polarization of the neutrophil phenotype (Figure 1, Table 1).

In a pioneer study, using three mouse tumor models: mesothelioma AB12, hybridoma, and Kras-derived lung cancer, the ability of transforming growth factor beta (TGF-β) to play a role in neutrophil polarization was demonstrated [14]. TGF-β inhibition with the small TGF-β type 1 receptor kinase (ALK5) inhibitor SM16 increased the levels of neutrophil chemoattractants in the tumor microenvironment, resulting in neutrophil recruitment [14]. In all tumor models, the gene expression profiles of TANs from SM16-treated tumors revealed a significant decrease in arginase levels and a significant increase in tumor necrosis factor alpha (TNF-α) and intercellular adhesion molecule 1 (ICAM1) levels compared with TANs from SM16-untreated mice [14]. Arginase overexpression could lead to L-arginine depletion in the tumor microenvironment, which impairs T cell function and supports tumor immune escape [18]. Elevated levels of TNF-α and ICAM1 indicate the pro-inflammatory status of TANs from SM16-treated tumors. Functional analysis revealed enhanced cytotoxicity of TANs isolated from SM16-treated tumors against tumor cells, while TANs from untreated tumors were found to be noncytotoxic. In mesothelioma AB12 tumors of SM16-treated mice, in vivo depletion of CD8+ T cells by mAb injection canceled the reduction in tumor growth, indicating a dependence of TAN anti-tumor effects on CD8+ T cells. In SM16-untreated mice, in vivo TAN depletion with or without CD8+ T cell depletion led to a significant decrease in tumor size, indicating the pro-tumor activities of TANs [14]. The findings of this study provide a basic understanding of the morphological and functional differences between neutrophil N1 and N2 phenotypes, which are primarily regulated by TGF-β.

**Figure 1 ijms-23-15827-f001:**
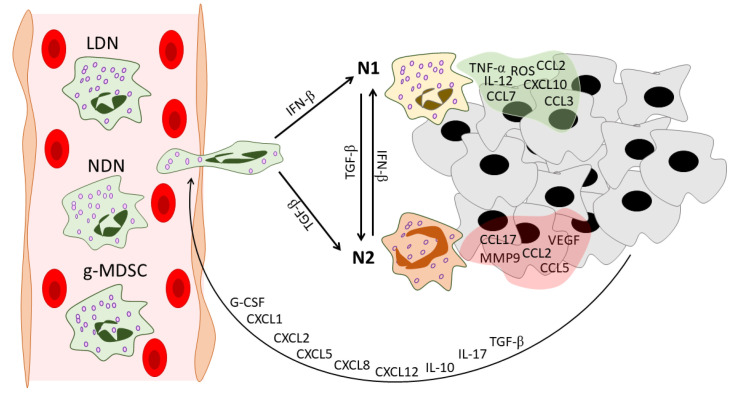
Neutrophil heterogeneity during tumor development. In the peripheral blood of cancer patients, three distinct populations of circulating neutrophils can be found: NDNs, LDNs, and g-MDSCs. Tumors recruit neutrophils via various mediators. These mediators include G-CSF [19], CXCL1 [20], CXCL2 [21], CXCL5 [22], CXCL8 [23], CXCL12 [24], IL-10 [19], IL-17 [25], and TGF-β [26]. After infiltration into the tumor microenvironment, neutrophils gain an N1 or N2 phenotype under the action of IFN-β [27] or TGF-β [14], respectively. Neutrophils in their turn reshape the tumor microenvironment: N1 TANs secrete pro-inflammatory anti-tumor mediators [14,28], while N2 TANs support tumor progression and angiogenesis and enhance the immunosuppressive tumor microenvironment [24,28]. NDNs—normal-density neutrophils, LDNs—low-density neutrophils, g-MDSCs—granulocytic-myeloid-derived suppressor cells, G-CSF—granulocyte colony-stimulating factor, CXCL—C-X-C motif chemokine ligand, CCL—C-C motif chemokine ligand, IL—interleukin, TGF-β—transforming growth factor beta, IFN-β—interferon beta, TNF-α—tumor necrosis factor alpha, ROS—reactive oxygen species, VEGF—vascular endothelial growth factor, MMP9—matrix metallopeptidase 9, TME—tumor microenvironment.

Later, interferon beta (IFN-β) was identified as the orchestrator of neutrophil polarization toward the N1 phenotype in cancer patients and tumor-bearing mice [24,27,29]. In *Ifnb1*^−/−^ mice after B16F10 melanoma implantation, enhanced tumor growth, angiogenesis, and metastasis were observed and accompanied by higher levels of TANs compared with tumors developed in *Ifnb1*^+/+^ mice. TANs isolated from *Ifnb1*^−/−^ mice (*Ifnb1*^−/−^-TANs) highly expressed C-X-C motif chemokine receptor 4 (CXCR4) and its regulators c-Myc and signal transducer and activator of transcription 3 (STAT3), vascular endothelial growth factor (VEGF), and matrix metallopeptidase (MMP9) [24]. CXCR4 traffics neutrophils via a gradient of CXCL12, which was overexpressed in the tumors of *Ifnb1^−^/^−^* mice compared to controls [24,30]. MMP9 is a proteolytic enzyme that degrades the ECM and paves the way for new vessels [31]. VEGF plays a well-known key role in angiogenesis and is an important suppressor of anti-tumor immunity in the tumor microenvironment [32,33,34]. Altogether, high expression of CXCR4, VEGF, and MMP9 could serve as an ideal triad for successful neutrophil-induced angiogenesis. Interestingly, in vitro treatment of *Ifnb1*^−/−^-TANs with exogenous IFN-β decreased the expression of the abovementioned genes [24]. This study sheds light on the regulatory role of IFN-β in the acquisition of pro-angiogenic properties by neutrophils.

The absence of IFN-β was also associated with a prolonged life span of blood neutrophils and TANs [29]. Pro-angiogenic TANs from *Ifnb*1^−/−^ mice were shown to have a prolonged life span in tumor-bearing mice, which could be explained by lower expression of FAS, active caspase 3 and 9, and an imbalance in the expression profiles of pro-apoptotic and anti-apoptotic genes [29]. Moreover, TANs from IFN-β–deficient mice showed a reduction in reactive oxygen species (ROS) production [29].

**Table 1 ijms-23-15827-t001:** Diverse neutrophil subpopulations in cancer in comparison with mature neutrophils in healthy individuals.

Neutrophil Type	Markers		Origin	Maturity	Location/ Detection	Life Span/ Turnover	ROS Production	Angiogenic Properties	NETosis	Interactions with Adaptive Immunity	Other Features
	Human	Murine									
**Mature neutrophils**	CD11b^+^ CD16^+^ CD15^+^ CD14^−^ [35,36]	CD11b^+^ Ly6G^+^ Ly6C^−^ [35,36]	Hematopoietic stem cells in bone marrow [3]	In the final steps of granulopoiesis, neutrophils gain morphological and surface markers of maturity [3]	Bone marrow, peripheral blood, spleen, and tissues [37]	In blood, neutrophils have half-lives of 12.5 h for mice and 90 h for humans [38]; in tissues, neutrophils undergo apoptosis or reverse migration [35]	At the site of infection, neutrophils release large amounts of ROS as an antimicrobial mechanism [39]	Neutrophils in tissues may exhibit a non-immune angiogenic phenotype [6]	Undergo NETosis in response to various microorganisms and endogenous stimuli [40]	Are involved in complex interactions, including the activation and regulation of other immune cells [41]	N.D.
**N1 TANs**	Carry markers similar to mature neutrophils		Can come from both LDNs and, most likely, NDNs in the blood and tumor microenvironment [42]	Mature cells [14]	Intratumoral [14]	N.D. Polarization to N1 by IFNs could delay neutrophil apoptosis [43,44]	Able to produce high levels of ROS [45]	IFN-β maintains the low levels of expression of angiogenic factors in N1 TANs [24]	Polarization to N1 by IFNs could ensure the capacity of N1 TANs to produce NETs [46]	Activate CD8^+^ T cells [14]; participate in antigen presentation [28]	Hyper-segmented nucleus [14]
**N2 TANs**	Carry markers similar to mature neutrophils		Can come from both NDNs and, most likely, LDNs [42]	Show morphological signs of immaturity [14,27]	Intratumoral [14]	N.D. Could have a prolonged life span [29]	Reduced [29]	Produce high levels of CXCR4, VEGF, and MMP9 [24]	Reduced [27]	Could recruit Tregs [28]; produce high levels of arginase [14]	Circular nucleus [14,27]
**LDN**	CD11b^+^ CD16^+^ CD15^+^ CD66^+^ Siglec8^-^ CD36^high^ CD61^high^ CD41^high^ Lox1^high^ CD226^high^ CD10 ^+/−^ [47]	CD11b^+^ Ly6G^+^ [15]	Could originate from NDNs under the action of tumor-derived factors [42]	Consist of both mature and immature populations [15]	Blood of cancer patients and tumor-bearing mice [15], could infiltrate tumors [42]	LDNs showed a lower rate of apoptosis in vitro in comparison to NDNs [15]	Increased [42]	N.D.	Immature LDNs in response to stimulation in vitro show increased ability to NETosis [48]	Express higher levels of PD-L1 in comparison to NDNs [49]	Lower phagocytic activity [42]; immature LDNs have greater bioenergetic capacity [48]
**g-MDSC**	CD11b^+^ CD15^+^ CD14^−^ CD66b^+^ CD33+ HLA-DR^-^ Lox1^+^ [19,50]	CD11b^+^ Ly6G^+^ Ly6C^low^ [50]	Granulocytic precursors [51]	Immature cells [35]	Bone marrow, blood, spleen, and tumors of tumor-bearing mice; blood and tumor environment of cancer patients [52]	N.D. Their turnover could be regulated by the Fas-FasL pathway [53]	Increased [54]	Could participate in tumor angiogenesis [55]	Could produce NETs under specific conditions [56]	Suppress T cells [57]	Lower density [58]; lower phagocytic activity [59]

Andzinski et al. clearly showed the ability of IFN-β to polarize neutrophils in anti-tumor phenotype [27]. In tumor-bearing mice, upon IFN-β deficiency, neutrophil turnover and mobilization were faster and were combined with a higher percentage of immature neutrophils with ring-shaped nuclei in the blood [27]. In a co-culture with tumor cells, TANs from IFN-β-deficient mice showed significantly lower cytotoxicity and TNF-α expression in comparison with TANs from wild-type mice. However, the anti-tumor cytotoxicity of TANs was recovered after adding exogenous IFN-β to the co-culture [27]. Thus, the phenotypic switch of neutrophils could be regulated by TGF-β and type 1 IFN antagonistic signaling pathways [60,61].

However, the fate of neutrophils to be friend or foe is probably decided by multiple factors, and not only in the tumor microenvironment but outside it. For example, Yan et al. showed that interleukin 6 (IL-6) along with granulocyte colony-stimulating factor (G-CSF) induces the neutrophil N2 phenotype in the bone marrow, a process most likely regulated by the immune suppressor cytokine IL-35 [62,63]. Moreover, it has also been suggested that neutrophils act differently depending on the stage of tumor development [64,65]. TANs isolated from early tumors produced higher levels of NO, H_2_O_2_, and TNF-α and demonstrated greater cytotoxicity against tumor cells in comparison with TANs isolated from late-stage tumors [64]. Interestingly, tumor growth was unaffected by neutrophil depletion during the early stages of tumor development. In contrast, after tumor establishment, neutrophil depletion led to a significant reduction in tumor growth, indicating a pro-tumorigenic effect of neutrophils at the late stage of tumor development [64]. Neutrophil migratory properties also vary in different stages of tumor development [65]. At early stages, neutrophils show enhanced migratory and metabolic potential with no immunosuppressive function. However, in later stages, neutrophils lose their elevated migratory and metabolic properties and gain an immunosuppressive phenotype [65].

Shaul et al. deeply analyzed the N1 and N2 phenotypes of neutrophils using microarray analysis and identified different transcriptomic signatures of N1 versus N2 neutrophils [28]. In the N1 profile, 136 genes were overexpressed and 2 genes were downregulated with a fold change of ≥10 [28]. N2 TANs showed a relative downregulation of genes associated with cytoskeletal organization and actin polymerization compared with bone marrow neutrophils and N1 TANs, suggesting that after neutrophil infiltration into the tumor, N2-polarized TANs lose the ability to organize the cytoskeleton and to leave the tumor microenvironment [28]. N1 TANs showed an upregulation of many genes associated with antigen presentation, especially major histocompatibility complex type 1 (MHC-I)-related loci. Moreover, many integrins and membrane receptors associated with neutrophil immune responses are overexpressed in N1 compared with N2 TANs. For example, IFN-γ receptor 1 is expressed in bone marrow naive neutrophils and N1 TANs but is significantly downregulated in N2 TANs, which may result in a loss of communication between neutrophils and IFN-γ-releasing cytotoxic T cells [28]. N1 TANs have pro-inflammatory properties with higher expression levels of the pro-inflammatory cytokines IL-12 and TNF-α as well as various chemokines that attract T cells and macrophages—C-X-C motif chemokine ligand 10 (CXCL10) and C-C motif chemokine ligands 2, 3, and 7 (CCL2, CCL3, and CCL7) [28]. CCL17, which recruits Tregs, is downregulated in N1 TANs compared to N2 TANs, another mechanism of the immunosuppressive function of N2 TANs [28].

Ohms et al. first polarized human neutrophils in vitro [45]. A cocktail containing lipopolysaccharide (LPS), IFN-γ, and IFN-β was used to polarize neutrophils toward an N1-like phenotype, while L-lactate, adenosine, TGF-β, IL10, prostaglandin E2 (PGE2), and G-CSF together were used to polarize neutrophils toward an N2-like phenotype [45]. Since neutrophils have a short life span and spontaneously undergo apoptosis, pan-caspase inhibitor was added during the polarization process [45]. The cytokine profile and functional features of in vitro-polarized neutrophils correspond to those of in vivo-polarized ones, allowing the investigation of deeply different phenotypes of neutrophils in vitro [45]. Lovászi et al. applied the protocol provided by Ohms et al. [45] to investigate the role of the neutrophilic A2A adenosine receptor (A2AAR) in neutrophil polarization [66]. A2AAR-specific agonist CGS21680 was added to the N1 polarization cocktail, and A2AAR-selective antagonist ZM241385 was added to neutrophils before adding the N2 polarization cocktail. The activation of A2AAR skewed N1 neutrophils to the N2 phenotype, while blocking A2AAR suppressed N2 polarization, which indicates the crucial role of the adenosine–A2AAR axis in N2 neutrophil polarization [66]. The discovery of the pro- and anti-inflammatory profiles of N1 and N2 neutrophils, respectively, has led to a wide investigation of these two phenotypes in several physiological and pathological conditions, including inflammatory diseases [67,68], bone regeneration [69], ischemia [70], myocardial infraction [71], and Alzheimer’s disease [72]. Of note, N1/N2 neutrophil classification in terms of infection could differ from N1/N2 TANs described in terms of tumor, which should be considered when moving from one research field to another. However, LPS-stimulated neutrophils showed a phenotype similar to that of anti-tumor N1 neutrophils, which may indicate a relationship between the pro-inflammatory and anti-tumor functions of neutrophils [73].

### 2.2. NDN vs. LDN

In differential density centrifugation, the main proportion of neutrophils is purified in a high-density layer and called high-density neutrophils (HDNs). However, a significant proportion of neutrophils were found to co-purify with the low-density mononuclear cell layer and are called low-density neutrophils (LDNs) [15] (Figure 1, Table 1). This heterogeneity in neutrophil density was described in 1983 [74]. To avoid confusion, since the term HDN does not refer to a specific neutrophil subpopulation except neutrophils with unaltered normal density, normal-density neutrophils (NDNs) seems to be a more suitable term [75], and thus we use it in this review. It should be noted that TANs can come from both NDNs and LDNs [42], but because LDNs are more likely to have a pro-tumor phenotype [76], we hypothesized that N1 TANs come from the NDN fraction and N2 TANs come from the LDN fraction after entering the tumor microenvironment from the bloodstream.

The elevated levels of LDNs in the blood of cancer patients and tumor-bearing mice resulted in the study of their functions and the molecular pathways involved in their elevation during cancer development [15,47,77,78]. Interestingly, TGF-β was also involved in NDN to LDN switching [15]. Guglietta et al. showed that NETosis-induced blood clots could also switch NDN to LDN and suggested, based on gene expression profiling, that LDNs have an intermediate profile between an NDN and N2 [79]. In comparison to NDNs, LDNs from cancer patients overexpress CD66b, CD11b, and CD15 [15,80]. Shaul et al. performed cytometry by time-of-flight (CyTOF) analysis of NDNs and LDNs from healthy individuals and patients with lung cancer. Their data showed significant differences in the expression of CD10, CXCR4, CD94, and programmed death-ligand 1 (PD-L1) between NDNs and LDNs. In both healthy individuals and cancer patients, two populations of NDNs were identified: CD66b^high^/CD10^high^/CXCR4^med^/PDL1^low^ and CD66b^high^/CD10^med^/CXCR4^med/low^/PDL1^low^ neutrophils. Heterogeneous subsets in the LDN fraction from cancer patients were demonstrated and a unique subset defined by CD66^high^/CD10^low^/CXCR4^high^/PDL-1^high/med^ was identified [78].

In patients with pancreatic ductal adenocarcinoma (PDAC), increased levels of circulating LDNs, which included cycling and non-cycling precursors, immature as well as mature neutrophils were observed [5]. The LDN fraction, isolated from the peripheral blood of stem cell donors receiving recombinant G-CSF, is composed of both immature (CD10−) and mature (CD10+) neutrophils [81]. Valadez-Cosmes et al. performed a high-dimensional screening of human cell surface markers and identified various markers that are overexpressed in LDNs which allowed them to discriminate between LDN and NDN subpopulations in cancer patients [47]. In the LDN subpopulation, the highest fold change was found in the CD36, CD41, CD61, and CD226 markers [47]. Functional analysis revealed impaired phagocytic activity, impaired ROS production, and the absence of anti-tumor activity in the LDN mature fraction, which corresponds to the results published by Marini et al. where mature (CD10+) LDNs inhibited T cell functions whereas immature (CD10−) LDNs enhanced them [15,81]. Furthermore, compared with NDNs, LDNs express higher levels of PD-L1 and can inhibit cytotoxic T cells and natural killer (NK) cells [49,82]. In a recent study, Arasanz et al. showed a possible role of circulating LDNs in the development of resistance to PD-1/PDL1 immunotherapy in non-small-cell lung cancer (NSCLC) patients [83]. In breast cancer patients, LDN levels were associated with a worse prognosis and were significantly higher in the case of metastatic cancer than in non-metastatic cases [77]. Similar results were observed in breast-cancer-bearing mice, where LDNs were involved in promoting liver metastasis [48]. In addition to studying the role of LDNs in cancer development, LDNs are actively investigated in inflammatory diseases [84], infections [85,86], and autoimmune diseases [87].

### 2.3. g-MDSCs

In the field of neutrophil heterogeneity, we should mention myeloid-derived suppressor cells (MDSCs) (Figure 1, Table 1). MDSCs are a population of immature myeloid cells derived from the granulocytic (g-MDSCs) or monocytic (m-MDSCs) lineages with a remarkable ability to suppress T cells [57]. MDSCs have been shown to accumulate in cancer patients and tumor-bearing mice and have also been observed under different conditions, including infection, chemotherapy, experimentally induced autoimmunity, and stress [88]. The similarity in the morphology and phenotype of g-MDSCs and mature neutrophils makes it difficult to distinguish between these populations [89].

In their recent review, Que et al. summarized the studies in which g-MDSCs were believed to be a neutrophil subset or a distinct population [90]. The authors described a TAN as a “similar entity” to a g-MDSC, which is a suitable description in this context [90]. From our viewpoint, this problem appears to be more relevant to the subject of nomenclature, and there is a need to standardize the nomenclature of different neutrophil populations. All in all, at present, the scientific community has adopted the concept of neutrophils as a heterogeneous population that exhibits antagonistic effects in health and disease, including immunosuppressive and pro-tumor ones.

## 3. Tumor–Neutrophil Crosstalk

### 3.1. Anti-Tumor Functions of Neutrophils

Although most recent studies focused on the pro-tumor effects of neutrophils, some studies have shown the ability of neutrophils to exhibit anti-tumor actions (Figure 2A). First, neutrophils exhibit direct anti-tumor activity via ROS and reactive nitrogen species (RNS) production (Figure 2(A1)). Using a mouse model of breast cancer, it was shown that in so-called pre-metastatic organs, including the lungs, neutrophil-derived H_2_O_2_ mediates tumor cell killing [91]. Gershkovitz et al. have shown that neutrophil-produced H_2_O_2_ increases Ca^2+^ concentrations in tumor cells to lethal levels [92]. Neutrophil-produced NO has also been shown to mediate tumor cell killing [93]. Tumor-derived factors induce the expression of the mesenchymal–epithelial transition tyrosine kinase receptor (MET) in neutrophils. MET interacts with its ligand hepatocyte growth factor (HGF) and leads to NO-mediated tumor cell killing [93]. In a mouse tumor model, it was shown that radiation therapy triggers the secretion of CXCL1, CXCL2, and CCL5, which leads to neutrophil recruitment to tumor sites; in turn, neutrophils generate ROS and suppress PI3K/AKT/SNAI1 signaling, inhibiting epithelial–mesenchymal transition [94]. Moreover, NETs could be included in neutrophil-mediated anti-tumor activities [95,96] (Figure 2(A1)).

Neutrophils can destroy cancer cells by antibody-dependent cellular cytotoxicity (ADCC) (Figure 2(A2)), firstly described by Erna Möller in 1965 [97,98]. In ADCC, antibodies bind to their specific antigens on tumor cells via Fab and then to the Fc-receptor on the immune effector cell via Fc, acting as a bridge between tumor and effector cells [99]. This assembly activates the effector cell, which then destroys the tumor cell [99]. Neutrophils express FcR on their surface, so they can be considered as potential effector cells for mAb-mediated tumor eradication [100]. One possible mechanism of neutrophil ADCC is the release of tumoricidal mediators by neutrophils after their interaction with mAb-coated tumor cells [90]. Recently, trogoptosis was suggested as a new mechanism of neutrophil ADCC [101]. Matlung et al. demonstrated that neutrophils could lyse tumor cells via an antibody-dependent repeated trogocytosis, referred to as trogoptosis [101]. Trogocytosis is the process when one cell “bites” and ingests small fragments of another cell [102]. It happens between two live cells and is believed to play a role in cellular communication [103]. During trogoptosis, neutrophils extensively eat small fragments of the tumor cellular membrane, leading to membrane destruction and necrotic cell death [101].

Neutrophils efficiently communicate with other immune cells and can modify the immune responses in the tumor microenvironment (Figure 2(A3)). Ponzetta et al. showed that neutrophils stimulate IL-12 production by macrophages, resulting in the polarization of CD4^−^CD8^−^ unconventional αβ T cells, which mediate IFN-γ-dependent immune resistance to 3-methylcholanthrene-induced sarcoma [104]. Additionally, TANs enhance the anti-tumor immunity by boosting CD8+ T cell reactivity to T cell receptor triggers [105]. Furthermore, ROS produced by TANs cause oxidative stress in IL-17–producing γδ T cells (γδ17 T cells) [106]. The inhibition of γδ17 T cells prevents the development of a pro-tumor immunosuppressive tumor microenvironment rich in IL-17.

### 3.2. Pro-Tumor Functions of Neutrophils

In addition to their anti-tumor functions, neutrophils can contribute to tumor initiation. Neutrophil-produced ROS cause DNA mutations and promote tumorigenesis in epithelial cells (Figure 2(B1)), as was shown by Knaapen et al. with the example of lung epithelial cells [107,108]. Neutrophils cause telomere DNA damage in hepatocytes, thereby enhancing hepatocellular carcinoma (HCC) development [109]. Neutrophil-derived ROS enhanced tumorigenesis in lung cells exposed to a carcinogen [110]. Not only neutrophil-produced ROS but also RNS can be genotoxic and are believed to contribute to tumorigenesis [111]. Recently, Butin-Israeli et al. showed that neutrophils in an inflamed colon could contribute to tumorigenesis through an ROS-independent mechanism, which includes the production of microvesicles loaded with pro-inflammatory miR-23a and miR-155 [112]. These miRNAs promoted the accumulation of double-strand DNA breaks by inducing the collapse of lamin B1-dependent replication forks, inhibition of homologous recombination, and impeding tissue healing [112].

Besides tumor initiation, neutrophils promote tumor growth through various direct and indirect mechanisms (Figure 2(B2–B4)). In several studies, PGE2 production by neutrophils was shown to enhance tumor cell growth [113,114,115]. In a co-culture model, neutrophil–A549 cell interaction resulted in the enhancement of tumor cell proliferation via the production of PGE2 and neutrophil elastase (NE) by neutrophils [115]. NE could penetrate into tumor cells and degrade insulin receptor substrate-1, which led to the enhancement of tumor cell proliferation via the PI3K axis [116]. Another mechanism is the production of IL-1 receptor antagonist in the tumor microenvironment, which supports malignant transformation and tumor growth [117]. Neutrophils enhanced the proliferation capacity of renal cell carcinoma cell lines via the induction of androgen receptor expression in cancer cells [118].

Neutrophils could create an immunosuppressive tumor microenvironment (Figure 2(B2)), primarily by the production of arginase [14,119], interleukin 10 (IL-10) [119], inducible nitric oxide synthase (iNOS) [119], and CCL17 [28,120]. In addition, TANs have been shown to express PD-L1 [82,121,122,123]. PD-L1 interacts with PD-1 on activated immune cells, especially activated T cells, inhibits T cell proliferation, and causes immune tolerance [124,125].

Neutrophils can contribute to tumor angiogenesis via the secretion of several angiogenic factors (Figure 2(B3)). Neutrophils could enhance tumor angiogenesis via the production of VEGF and MMP9 [24]. Galdiero et al. have shown that neutrophils secrete CXCL8/IL-8, VEGF-A, and MMP9 in response to the conditioned media of tumor cells [126]. Furthermore, TANs secrete HGF and high levels of prokineticin 2 (Bv8), a potent mitogenic factor for endothelial cells and the main angiogenic factor in neutrophil-dependent tumor angiogenesis [127,128,129,130,131]. Massena et al. have proved the presence of VEGFR1 (VEGF receptor 1) on both human and mouse neutrophils [132]. A unique subset of neutrophils, CD49d^+^VEGFR1^high^CXCR4^high^, was shown to intensively infiltrate into the hypoxic region via the VEGF-A/VEGFR1 axis and promote angiogenesis [132]. Besides angiogenesis, tumor cells could use pre-existing blood vessels to support their growth, a process called vessel co-option [133]. Another non-angiogenic vascularization mechanism is vascular mimicry in which tumors try to mimic normal vessels and build cancer-cell-based vascularization [134]. Interestingly, recent studies suggested a possible role for neutrophils in both vessel co-option and vascular mimicry [135,136].

Neutrophils can support cancer metastasis in several ways (Figure 2(B4)). Neutrophils secrete high levels of TNF-α and TGF-β, which significantly stimulate tumor cell migration and invasion [137]. In response to tumor-derived granulocyte–macrophage colony-stimulating factor (GM-CSF), neutrophils secrete high levels of transferrin (TRF), an iron transporter and potent mitogen, thus enhancing tumor growth and metastasis [138]. In breast cancer models, Li et al. found that neutrophils could supply tumor cells in the premetastatic niche with their own lipids in a macropinocytosis–lysosome pathway, thus supporting tumor growth [139]. Bellomo et al. showed that neutrophils could support metastatic PDAC cells in the liver after chemotherapy via the production of growth-arrest-specific protein 6 (Gas6), which interacts with its receptor AXL on tumor cells, activating tumor cells and mediating metastatic relapse [140]. Moreover, neutrophil interaction with circulating tumor cells (CTCs), which are responsible for the development of metastasis in several types of cancer, could enhance the efficacy of CTCs to develop metastases [141].

### 3.3. Tumor Cells Skew Neutrophils toward a Pro-Tumor Phenotype

Recent studies have demonstrated the ability of tumor cells to promote pro-tumorigenic neutrophil functions. This potential to polarize neutrophils is realized by different ways of intercellular communication, mainly by the secretion of various soluble mediators and extracellular vesicles (Figure 3). For example, Anselmi et al. showed that human melanoma stem cells can activate and polarize neutrophil-like HL-60 cells toward the N2 phenotype via the production of polarizing factors such as TGF-β, IL-8, and IL-6 [142]. Niu et al. showed that breast cancer cells modify neutrophils to an immunosuppressive phenotype by secreting serum amyloid A 1 (SAA1) [119]. SAA1 interacts with toll-like receptor 2 (TLR2) on neutrophils and promotes neutrophils to produce IL-10, arginase, and iNOS, indicating the immunosuppressive activities of SAA1-treated neutrophils. In the 4T1 breast cancer model, neutrophil blocking with anti-LY6G mAb with or without anti-SAA1 mAb slowed tumor growth. The slowest tumor growth rate was observed when a combination of anti-LY6G and anti-SAA1 mAbs were used together [119].

**Figure 2 ijms-23-15827-f002:**
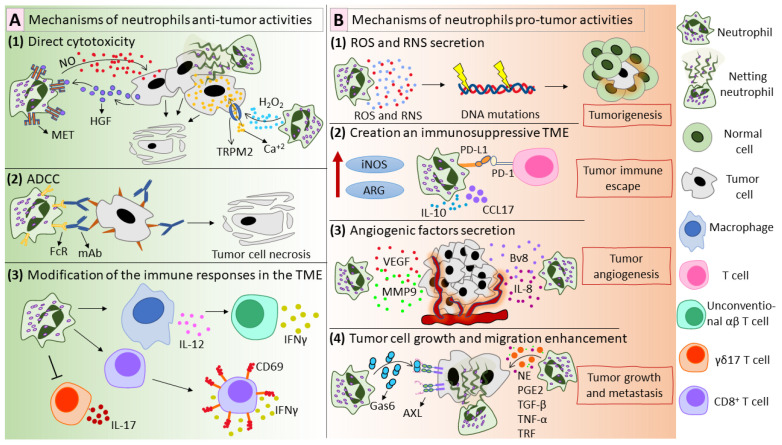
Mechanisms of anti-tumor (**A**) and pro-tumor (**B**) activities of neutrophils. (**A**) Mechanisms of neutrophil anti-tumor activities. (**A1**) Neutrophils exhibit direct anti-tumor activity via the production of ROS and RNS. Neutrophil-derived H_2_O_2_ activates TRPM2 and kills tumor cells in a CA^+2^-dependent manner [92]. Tumor-derived HGF interacts with MET on neutrophils and stimulates NO production, which mediates the destruction of tumor cells [93]. Moreover, NETs can display anti-tumor effects [95,96]. (**A2**) Neutrophils kill antibody-coated tumor cells via ADCC in a mechanism called trogoptosis [101]. (**A3**) Neutrophils alter the immune responses in the tumor microenvironment. Neutrophils stimulate macrophages to produce IL-12, which leads to the polarization of CD4^−^CD8^−^ unconventional αβ T cells, which exhibit IFN-γ-dependent anti-tumor activity [104]. Moreover, neutrophils enhance CD8+ T cell reactivity, reflected in CD69 expression and IFN-γ secretion and inhibit γδ17 T cells [105,106]. (**B**) Mechanisms of neutrophil pro-tumor activities. (**B1**) Neutrophils produce ROS and RNS, which can cause genotoxicity and contribute to tumorigenesis [110,111]. (**B2**) Neutrophils participate in creating an immunosuppressive tumor microenvironment by expressing PD-L1 on their surface, producing high levels of iNOS and ARG and secreting immunosuppressive mediators such as CCL17 and IL-10 [28,119,120,121]. (**B3**) Neutrophils support tumor angiogenesis via the secretion of several factors: VEGF [24], MMP9 [24], IL-8 [126], and Bv8 [127]. (**B4**) Neutrophils promote tumor growth and metastasis by producing NE [116], PGE2 [115], TGF-β [137], TNF-α [137], TRF [138], Gas6 [140], and NETs [143]. ROS—reactive oxygen species; RNS—reactive nitrogen species; TRPM2—transient receptor potential cation channel, subfamily M, member 2; HGF—hepatocyte growth factor; MET—mesenchymal–epithelial transition tyrosine kinase receptor; NETs—neutrophil extracellular traps; ADCC—antibody-dependent cellular cytotoxicity; IL—interleukin; IFN—interferon; PD-L1—programmed death-ligand 1; iNOS—inducible nitric oxide synthase; ARG—arginase; CCL17—C-C motif chemokine ligand 17; VEGF—vascular endothelial growth factor; MMP9—matrix metallopeptidase 9; Bv8—prokineticin 2; NE—neutrophil elastase; PGE2—prostaglandin E2; TGF-β—transforming growth factor beta; TNF-α—tumor necrosis factor alpha; TRF—transferrin; Gas6—growth arrest specific 6.

Tumor-produced CXCR2 agonists were also involved in the modification of neutrophil function. Safarulla et al. showed that brain-metastatic breast cancer cells (MDA-MB231BrM2a) modulate neutrophil function, most likely by the secretion of CXCR2 ligands (particularly CXCL1) [144]. MDA-MB231BrM2a-conditioned media enhanced neutrophil CXCR2 expression, increased neutrophil chemotaxis, and induced pro-metastatic NET production [144]. Moreover, tumor-secreted CXCL1 contributes to the modification of neutrophil behavior and drives NDNs to an LDN-like phenotype [42]. In the 4TO7-Lin28B breast cancer model, tumor cells induce the production of IL6 and IL10 and trigger the neutrophil polarization toward a pro-tumor phenotype, in which the expression levels of arginase 1, CD206, Ym1, IL-4, IL-10, TGF-β, and IL-6 were increased while the expression levels of TNF-α, iNOS, IL-12a, and IL-1β were decreased [145]. In addition, N2-converted neutrophils inhibited T cell proliferation, activation, and differentiation [145].

Interestingly, in addition to reprogramming neutrophils to a tumor-supportive state, cancer cells also promote neutrophil viability, presumably to benefit from their pro-tumor function as long as possible [146].

Overall, tumors reinforce neutrophils to act in their benefit. Different tumor types could control the tumor–neutrophil communication via different mediators, activating distinct molecular pathways in neutrophils.

### 3.4. Tumor Cells and NETs

Neutrophil extracellular traps, formed as the result of specific neutrophil death, NETosis, are web-like structures of neutrophilic origin containing both genomic material and various neutrophil granule proteins, including enzymes with lytic activity [12,147]. NETosis occurs in response to a variety of stimuli, including activators of sterile or infectious inflammation. NETs are supposed to be the main immune response to infection [148]. NETosis can be divided into at least two different types: lytic NETosis (activated through the Raf/MEK/ERK cascade followed by the production of ROS by NADPH oxidase), which results in the rupture of the cell membrane and neutrophil death, and vital NETosis (through TLR2/TLR4 receptors and Ca^2+^-dependent mechanisms), which results in the formation of a phagocytic nuclear-free cytoplast [149]. Some researchers have identified another type of NETosis: mitochondria-specific vital NETosis, in which the DNA core of NETs includes DNA of mitochondrial origin [150]. A detailed up-to-date summary can be found in the review by Huang et al. [151].

Neutrophils in the tumor microenvironment are actively exposed to NETosis [152,153,154,155]. In a co-culture with tumor cells or under the influence of the tumor-cell-conditioned medium, the neutrophil apoptosis rate is decreased, thus increasing the probability of the initiation of NETosis [156]. Tumor hypoxia may be one of the possible factors contributing to the increased NET production [157]. In addition, tumor cells and cells of the tumor microenvironment can secrete various factors (pro-inflammatory cytokines, proteases, and exosomes) that activate neutrophils and trigger NETosis (Table 2, Figure 3). To date, increased NET production has been observed in many types of cancer: breast cancer [158], gastric cancer [159], PDAC [160], diffuse large B-cell lymphoma [161], lung cancer [162], and small intestine cancer [157]. Not only tumor cells but also cancer-associated fibroblasts (CAFs) can enhance NETosis. CAFs can secrete a variety of cytokines (CXCL5 [163], CXCL6 [164], and IL-8 [165]) which change the profile of neutrophils and stimulate NETosis. In addition, CAFs can secrete amyloid β, a possible trigger of NETosis [166].

In turn, NETs can enhance carcinogenesis and support tumor growth. NETs contain many active factors such as myeloperoxidase (MPO), NE and other ROS producers, cathepsins, nuclear proteins such as high-mobility group protein B1 (HMGB1), interleukins, and other components [167]. Most components of NETs are capable of activating tumor cells. For example, HMGB1 acts through TLR4/TLR9 receptors and activates tumor cells via the p38/NFkB pathway [167]. In addition, ROS cause DNA damage and genetic instability, thereby promoting carcinogenesis [168]. Moreover, NET-derived NE was shown to enhance tumor cell growth and mitochondrial biogenesis via the interaction with TLR4 on cancer cells [169]. In the Kras-driven pancreatic adenocarcinoma model, NETs enhanced tumor growth via the activation of pancreatic stellate cells [170]. Some tumor cells can be activated directly via CCDC25—integrated in the DNA core of NETs—by the activation of the β-parvin/Rac1/CDC42 pathway [171]. Moreover, through TLR4 signaling, NETs contribute to the differentiation and growth of Tregs, developing an immunosuppressive environment and facilitating the development of HCC [172]. Furthermore, NETs are also involved in dormant cancer cell activation. In mouse models, two main NET components were essential in this process: MMP9 and NE [173]. MMP9 and NE remodel laminin, which in turn activates dormant cancer cells [173]. Interestingly, NET-DNA serves as a scaffold, which binds to laminin and supports the proximity between NET-proteases and cancer cells [173].

NETs can also participate in tumor metastasis and angiogenesis. NET proteolytic enzymes can degrade the metastasis-suppressing protein thrombospondin-1 (THBS1), thus promoting metastasis [174]. Moreover, NET components degrade the extracellular matrix, extending the tumor microenvironment and facilitating the formation of the pre-metastatic niche [175,176]. Factors such as MMP9 in NETs may also contribute to tumor angiogenesis [177].

NETs may also promote the survival of CTCs. NETs can capture tumor cells to form pseudometastatic clusters and facilitate the extravasation and implantation of tumor cells in the liver and peritoneum [178,179]. Captured tumor cells by NETs are effectively hidden from effector immune cells [180,181].

In addition to direct interactions, NETs can be involved in more complicated interactions within the tumor microenvironment and contribute to the occurrence of systemic pathological processes. Many components of NETs can activate platelets and promote the coagulation cascade, contributing to the development of cancer-associated thrombosis. NETs promote platelet aggregation mainly due to negatively charged DNA and histones [182]. Moreover, both tissue factor (TF) and factor XII could be present in NETs [183,184,185]. Excessive NETosis can also lead to complications that are not directly related to tumor growth but often occur in cancer patients, such as systemic inflammation in organs other than the tumor host organ or metastatic sites, such as the heart and kidneys [186]. Additional risks associated with high levels of NETs in cancer patients include the high possibility of metastasis after tumor resection [148].

Interestingly, major components of NETs can theoretically display oncolytic properties. Indeed, in vitro newly formed NETs induced apoptosis in Caco-2, AML, and melanoma cells [187,188]. In the CT-26 mouse intestinal adenocarcinoma model, oncolytic vesicular stomatitis virus triggered an inflammatory response that included neutrophil-dependent thrombosis in tumor neovasculature, possibly mediated by NETs, which resulted in the suppression of tumor growth [95]. Neutrophils from patients with head and neck squamous cell carcinoma showed high cytotoxicity against tumor cells realized, most likely, due to NET production [96]. Liu et al. have shown that Bacillus Calmette–Guerin (BCG) treatment activates tumor cells, which produce TNF-α and IL-8, thus promoting the formation of NETs [189]. The anti-tumor effect of NETs was mainly realized by the induction of tumor cell apoptosis and cell cycle arrest. Furthermore, NETs stimulated CD3^+^ and CD14^+^ cell infiltration into the tumor, indicating that NETs can boost anti-tumor immune responses in the tumor microenvironment through the induction of T cells and macrophage infiltration [189].

### 3.5. Tumor Cells and Neutrophils Exchange Extracellular Vesicles

Extracellular vesicles (EVs) are membrane particles secreted by almost all cells. EVs are very heterogeneous in size, structure, content, and biogenesis. They transfer biological information between different cells, facilitating intercellular communication [190]. EVs generally include transmembrane and cytosolic proteins and peptides, lipids and their metabolites, miRNAs and mRNAs, and according to some data, genomic and mitochondrial DNA [191]. EVs are divided according to the type of biogenesis into microvesicles and exosomes; microvesicles are formed by budding of the plasma membrane and bear certain markers of the endoplasmic membrane; and exosomes are formed within the lumen of multivesicular endosomes, which fuse with the plasma membrane, releasing the exosomes into the intercellular space, and thus the exosomes bear endosomal markers [191]. Microvesicles are generally larger and have a diameter of 100 nm to 1 µm, while exosomes are smaller, measuring 30–150 nm in diameter. Microvesicles are additionally divided into microvesicles of large or medium sizes, including a wide range of heterogeneous vesicles such as apoptotic vesicles [192]. In addition, many researchers define the type of EVs depending on the cell of origin (oncosomes from tumor cells, prostasomes from prostate cancer cells) or by the type of biological process leading to the formation of vesicles (migrasomes which are produced during cell migration, apoptosomes which are produced during cell apoptosis, etc.) [191].

**Table 2 ijms-23-15827-t002:** Tumor-derived factors inducing NETosis.

Factor/Tumor-Associated Condition	Source	Possible Mechanisms of Action	References
Amyloid β	CAFs [166]	Directs the formation of tumor-associated NETs via CD11b and ROS-dependent mechanism	[166]
Cathepsin C	Tumor cells [174]	Induces neutrophil recruitment and NET production via the PR3-IL-1β-NF-κB axis	[174]
Complement component C3a	C3 proteolysis in the extracellular environment, tumor cells [193]	Causes neutrophils recruitment to the tumor microenvironment as well as LDN recruitment to liver metastasis sites	[193,194]
CXCL5	CAFs [163] and tumor cells [195]	Enhances neutrophil chemotaxis via the ERK/p38 pathway	[196,197]
CXCL6/ huGCP-2	Tumor cells [164]	Chemoattractant for neutrophils; NETosis induction via CXCR2	[198]
EVs	CAFs [199] and tumor cells [200]	Chemoattractant for neutrophils; depending on the load, they can change the phenotype of neutrophils; trigger NETosis, NF-κB pathway activation	[145,200]
G-CSF	CAFs and tumor cells [166]	NETosis induction via NOX, MPO, and ROS	[200,201]
Hypoxia		Regulation of NET release via the mTOR pathway and increased HIF-1α expression	[157]
Il-8	Tumor cells [156], endothelial cells [202]	Chemoattractant for neutrophils; induction of NETosis through CXCR2, followed by activation of the PI3K/p38/NF-kB pathway; possible induction via NOX, MPO, and ROS	[165,203,204,205]
IL-17	Th17, CD4+, and γδ T cells [206]	Mediates recruitment of neutrophils via CD8+ T cells, IL-17-Th (mechanism unknown)	[206]

A complete list of recommended markers for the classification of exosomes and microvesicles, as well as physical methods for their characterization, can be found in the guidelines by Théry et al. [207]. In this chapter of the review, we will use the general term extracellular vesicles (EVs).

Neutrophils, like many other cells, can produce EVs. In the 1970s, neutrophils were believed to produce ectosomes [208]. Much later, other types of extracellular vesicles of neutrophilic origin were characterized [209]. Activated neutrophils can form microvesicles ranging in size from 0.2 to 1 µm containing nucleic acids, proteins, and other molecules [210,211]. The composition of neutrophil EVs can differ according to the stimuli that enhance their formation. Moreover, microvesicles produced by attached neutrophils differ in composition and function compared to microvesicles from neutrophils in suspension [212].

It has been shown that neutrophils are capable of secreting small extracellular vesicles, called NEVs or NEX in different studies [213]. NEV biogenesis depends on neutrophil secretory granules. NEVs typically carry CD66b, CD11b, CD18, and MPO on their surfaces and can also expose phosphatidyl serine (PS) residues [214]. A detailed description of neutrophil microvesicles and other polymorphonuclear leukocytes can be found in the reviews [214,215].

Despite great interest in TAN-derived microvesicles, there are relatively few studies on the pro- or anti-tumor effects of NEVs. Some authors suggest that N1 and N2 neutrophils are able to release microvesicles with anti-tumor or pro-tumor properties according to the phenotype of the parent cell. Rubenich et al. speculated in their review the possible cargo and functions of NEVs from neutrophils of different phenotypes [213].

Rubenich et al. hypothesized that microvesicles produced by neutrophils with pro- or anti-inflammatory phenotypes could be similar to microvesicles produced by N1 or N2 TANs [213]. The candidate molecules for N1 microvesicles cargo are miR-223, 5-LO, FLAP, LTB4, S100A8/9, leukotrienes, integrins, and other pro-inflammatory cytokines, whereas N2 microvesicles could contain MMP2/9, CD66c, oncostatin M, defensin 1, IL-6, and S100A8/9. N1-derived NEVs could contain components with a dual effect, such as miR-223, which could act as a tumor suppressor or promoter [216].

The possibility of using neutrophils as a source of anti-tumor vesicles is attractive. A recent study showed that NEVs from human peripheral blood neutrophils can induce apoptosis in tumor cells [217]. However, the mechanisms behind these phenomena remain to be explored. Since neutrophil heterogeneity was proven in recent studies, the pool of released NEVs is also believed to be heterogeneous [214].

The functions of NEVs released by N2 pro-tumor neutrophils remain unclear. Some speculations contend that N2-derived NEVs could have pro-tumor properties [213]. In 4T1 and Met1 tumor models, the administration of NEVs derived from neutrophils after their stimulation with the cholesterol metabolite 27-hydroxycholesterol was shown to increase tumorigenicity and metastatic burden in mice [218]. This was realized by the effect of NEVs on myeloid cells, since NEVs were actively absorbed by macrophages as well as by neutrophils themselves but not by T cells [218].

Tumor-derived EVs (TEVs) are used by tumor cells to communicate with neutrophils (Figure 3). TEVs, which are much better studied than neutrophil-derived extracellular vesicles, play an important role in modulating the tumor microenvironment. Tumors are characterized by increased expression of microvesicle secretion regulatory factors (e.g., RAB27A), which is associated with a poor prognosis in pancreatic cancer and hepatocellular carcinoma [219,220]. Not only primary node tumor cells but also CTCs secrete TEVs [221]. Recipients of TEVs can be both cells of the proximal environment (fibroblasts) and immune cells, including macrophages [222], dendritic cells [223], T cells [224], NK cells [225], and neutrophils [200,226,227,228].

Neutrophils treated with TEVs acquired pronounced pro-tumor properties and activated the migration and proliferation of tumor cells [228]. It has been shown that TEVs of various origins can suppress neutrophil spontaneous apoptosis and increase their viability; this was shown for EVs derived from human breast carcinoma (MDA-MB-231 cells) [226], gastric cancer (BGC-823, MGC80-3, SGC-7901, and HGC-27 cells) [228], and melanoma cells (MV3) [73]. At the same time, EVs from non-tumor breast epithelial cells MCF10 and melanocytes NGM did not affect neutrophil viability [73,226]. Interestingly, pre-treatment of MDA-MB-231-derived EVs with annexin V led to inhibition of the stimulatory effect, indicating the importance of PS in these processes [226]. TEVs can also induce NETosis and NET release; however, it is not clear whether this NETosis is vital or lytic [73,226]. Additionally, gastric cancer cell-derived EVs increase autophagy in neutrophils [228]. TEVs may be a chemotactic agent for neutrophils since neutrophils migrate more actively toward MDA-MB-231-derived EVs but not toward EVs produced by non-tumor MCF10 epithelial cells [226].

TEVs promote the polarization of neutrophils into an N2-like phenotype (Figure 3). The absorption of the EVs—produced by breast carcinoma cells—by neutrophils increases the expression of IL-8, VEGF, arginase 1, MMP9, and CXCR4 (CD184), which are the main markers for N2 neutrophils [226]. Treatment of neutrophils with gastric cancer cell-derived TEVs enhanced the expression of MMP9 and VEGF [228]. It was shown that neutrophil activation under the action of gastric cancer cell-derived EVs occurs via the NF-κB pathway, and at the same time, an increase in the levels of p-p65, p-STAT3, p-ERK, and phosphorylated p-p38 and p-Akt was observed [228]. Treatment with an NF-κB inhibitor blocked TEV-induced STAT3 and ERK activation in neutrophils, increased spontaneous apoptosis, and decreased the expression of inflammatory factors [228]. In neutrophils treated with MV3-melanoma-derived EVs, enhanced gene expression of the N2 molecular markers arginase 1, CXCR4, VEGF, and CCL2 was observed, accompanied with a reduced expression of ICAM1 [73]. There was also an increase in phosphorylation of AKT, which suggests that N2 polarization occurs along the PI3K-AKT pathway [73]. An additional effect of TEVs is the stimulation of ROS release [73,226]. Breast carcinoma cell-derived EVs stimulate the production of ROS but not NO, whose production, in contrast, is reduced [226].

Of particular interest is the content of TEVs responsible for neutrophil activation. Zhang et al. showed that vesicular HMGB1 could be an active molecule in gastric cancer cell-derived EVs [228]. The activating effects of TEVs were reversed by treatment of vesicles with proteases and by pre-treatment of neutrophils with an HMGB1 antagonist, TLR4 inhibitors (but not TLR2 and RAGE inhibitors), and HMGB1 knockdown in tumor cells, suggesting the participation of the HMGB1/TLR4 axis in the observed effects of gastric cancer TEVs on neutrophils [228]. Leal et al. showed that 4T1 breast cancer cells produce EVs, which induce NET formation in neutrophils derived from G-CSF-treated mice [200].

Currently, there are many gaps in the understanding of the EV-dependent neutrophil–tumor communication, an interesting research area that will hopefully receive more attention in the near future.

**Figure 3 ijms-23-15827-f003:**
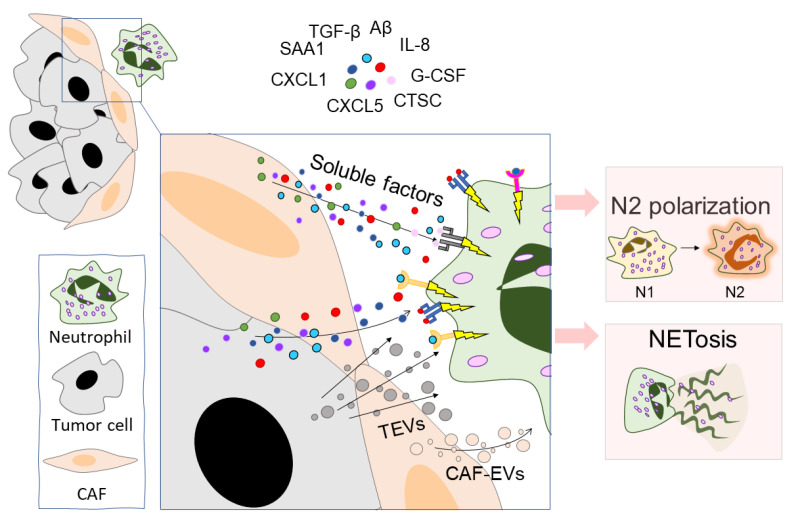
Tumor cells and CAFs modulate neutrophil function. Tumor cells and CAFs communicate with neutrophils through the production of EVs and several soluble factors. The main soluble factors are: TGF-β [14], SAA1 [119], CXCL1 [144], CXCL5 [163], IL-8 [165], Aβ [166], CTSC [174], and G-CSF [200]. The main effects of this communication are the polarization of neutrophils into the N2 phenotype and triggering NETosis. CAFs—cancer-associated fibroblasts, TGF-β—transforming growth factor beta, SAA1—serum amyloid A 1, CXCL—C-X-C motif chemokine ligand, IL-8—interleukin 8, Aβ—amyloid β, CTSC—cathepsin C, G-CSF—granulocyte colony-stimulating factor, TEVs—tumor-derived extracellular vesicles, CAF-EVs—CAF-derived extracellular vesicles.

## 4. Neutrophil in Cancer Therapy: Potential Strategies

Cancer therapies have achieved much in the past few decades. Cancer cells, however, are extremely adept at evading the immune system and developing resistance to therapy [229,230]. Since neutrophils can promote tumorigenesis and contribute to therapy resistance, targeting them may be considered a novel therapeutic approach in addition to standard therapeutic protocols [231]. In tumor-bearing mice, neutrophil depletion was shown to reduce tumor growth, decrease metastasis potency, and enhance immunotherapy efficiency [232,233,234]. Understanding neutrophilic phenotypes in cancer makes the possibility of neutrophil reprogramming or skewing toward an anti-tumor phenotype a potential therapeutic option. Additionally, several strategies affecting TANs and g-MDSCs were considered under preclinical and clinical conditions. These strategies include the inhibition of neutrophils’ pro-tumor activities or promoting their anti-tumor ones, inhibition of neutrophil recruitment to the tumor microenvironment, and targeting NETs. Moreover, neutrophils gained huge interest in the field of cancer immunotherapy, which exceeded expectations and led to the generation of CAR-neutrophils [17].

In this section, we try to summarize the potential therapeutic approaches that target neutrophil–cancer interplay (Figure 4). However, many other options are included in this field and have been well summarized in several recent reviews [10,90,235].

### 4.1. TGF-β Inhibitors

TGF-β, in addition to being the main cytokine involved in N2 polarization, is overexpressed in cancer cells and is central to cancer progression and immunosuppression [14,242]. Furthermore, TGF-β, produced by tumor cells, can act as a chemoattractant for neutrophils in the tumor microenvironment [26,243,244]. All this makes TGF-β an emerging target in cancer therapy [245]. De los Reyes et al. developed a mathematical approach to optimize the N2 to N1 transition in cancer patients using TGF-β inhibitors and IFN-β, and their results were promising enough to be used in clinical research [246]. Several in vitro and in vivo studies supported the mathematical models and demonstrated the potential of modulating the neutrophil functional state or inhibiting neutrophil recruitment to the tumor microenvironment using TGF-β inhibitors [243,247,248,249]. Qin et al. found that neutralizing TGF-β with monoclonal antibodies in a co-culture of primed neutrophils and SW480 cells (colon adenocarcinoma cells) inhibited tumor cell migration and increased neutrophil cytotoxicity against tumor cells. In vivo experiments revealed that anti-TGF-β antibodies retarded tumor growth in tumor-bearing mice compared with control tumors, an effect that was lost when neutrophils were depleted [247]. Jackstadt and colleagues showed that TGF-β promotes neutrophil recruitment to metastatic sites in a mouse model of metastatic colorectal cancer [243]. Inhibiting the TGF-β signaling pathway, either with a TGF-β-neutralizing antibody or an ALK5 inhibitor, resulted in a significant reduction in the number of neutrophils in metastatic sites, which was combined with a markedly reduced metastasis, indicating a significant contribution of TGF-β signaling to metastasis development in the liver by recruiting neutrophils [243,248].

Galunisertib, an ALK5 inhibitor, showed promising results in in vitro and in vivo cancer studies [250,251], passed the preclinical investigation [252], and has already been enrolled in several clinical studies for cancer management [253,254,255,256,257,258,259,260]. LY2109761, a TGF-β receptor type 1 and 2 dual inhibitor, has also been studied in human and murine tumor mouse models [261,262]. LY2109761 showed good results in sensitizing cancer cells to radiotherapy in in vitro and in vivo experiments [263]. LY2109761, in combination with oxaliplatin, a conventional chemotherapeutic agent, inhibited tumor growth and metastasis by enhancing anti-tumor immunity. Moreover, LY2109761 inhibits MDSC tumor infiltration [249]. In addition, LY2109761 improved the effect of transarterial embolization in a liver cancer animal model [264]. All of the above suggest LY2109761 as a potential agent in cancer combination therapy; however, more investigations are needed to move to clinical studies.

### 4.2. CXCR2 Axis Blockade

Another possible strategy to suppress neutrophil pro-tumorigenic effects is to inhibit neutrophil recruitment to the tumor microenvironment (Figure 4). Neutrophil recruitment to the tumor microenvironment is achieved by several factors that are produced by cancer and cancer-associated stromal cells [236,265]. Cancer-cell-produced agonists to CXCR2 are widely involved in neutrophil recruitment to the tumor microenvironment [21,25,26,165,266,267,268]. In addition, the CXCLs–CXCR2 axis has gained interest in clinical conditions. CXCR2 overexpression in human lung cancer tissue has been linked to a poor prognosis, while CXCR2 agonists have been proposed as potential diagnostic biomarkers in pancreatic cancer patients [269,270]. Moreover, CXCR2 agonists are involved in tumor-supporting NET production [144,180]. In addition, CXCR2 ligands were involved in neutrophils’ excessive biological aging, which promotes a more pro-tumorigenic state in neutrophils [271]. However, the controversial role of neutrophils in cancer progression is problematic for this approach which therefore benefits from the dedicated presence of pro-tumorigenic TANs.

Triple-negative breast cancer cells (TNBC) are known to recruit neutrophils via the production of huge quantities of CXCR2 ligands and TGF-β [26]. They can also polarize neutrophils to a pro-tumorigenic phenotype [26,226]. CXCR2 upregulation was found in TNBC themselves and was suggested as a novel cancer stem-like cell marker for TNBC [272,273]. Ghallab et al. showed that CXCR2 inhibition with the small-molecule inhibitor AZD5069 in TNBC culture eliminated doxorubicin resistance and improved the efficacy of atezolizumab, a monoclonal antibody against PD-L1 [273]. In the HCC with non-alcoholic steatohepatitis (NASH-HCC) mouse model resistant to anti-PD-1, AZD5069/anti-PD-1 combination therapy suppressed the tumor burden and extended survival [274]. Interestingly, combination therapy could modulate the phenotype of tumor-infiltrating neutrophils to an anti-tumorigenic one [274]. These findings could explain the anti-tumorigenic effect of AZD5069 despite its infectiveness in preventing neutrophil recruitment to tumors at the late stages of tumor growth [275]. AZD5069 was included in three clinical trials on cancer. In two studies, AZD5069 was used in combination with an anti-PD-L1 monoclonal antibody, durvalumab, and showed good results in breast and prostate cancer (NCT02499328). In another study, AZD5069 was used in combination with nonsteroidal hormonal antiandrogen therapy, enzalutamide, for patients with metastatic castration-resistant prostate cancer (NCT03177187).

Yang et al. investigated another selective CXCR1/2 inhibitor, SX-682, in tumor-bearing mice [276]. Their results showed that SX-682 administration alone or with anti-PD-1 monoclonal antibodies reduced the tumor burden, and the combination was significantly more efficient in comparison to vehicle control and to SX-682 or anti-PD-1 monotherapies [276]. In murine models of breast cancer, administration of SX-682 and/or bintrafusp alfa, an anti-PD-L1/TGF-β receptor II fusion protein, has shown moderate efficacy in slowing tumor growth [277]. In murine models of lung cancer, SX-682 or bintrafusp alfa monotherapies had no effect on tumor growth, and only the combinational therapy demonstrated a delay in tumor growth [277]. In short, combination therapy increased T cell and decreased g-MDSC infiltration into tumors and increased the epithelial phenotype of cancer cells [277]. Similar results were obtained in Sun et al.’s study, where SX-682 administration inhibited g-MDSC infiltration, promoted T cell accumulation in the tumor, and enhanced the effect of anti-PD-1 therapy or adoptive cell transfer of engineered T cell therapy in murine models of oral and lung cancer [278]. SX-682 also showed promising results in head and neck cancer models [279]. SX-682 is currently being investigated in different clinical trials in combination with anti-PD-1 monoclonal antibodies for melanoma stages III and IV (NCT03161431), for different types of colorectal cancer (NCT04599140), and for metastatic PDAC (NCT04477343). SX-682 monotherapy is also being investigated in myelodysplastic syndromes (NCT04245397).

### 4.3. Targeting Neutrophils to Restore the Efficiency of Immune Checkpoint Inhibitors 

One of the key mechanisms of tumor-induced immune suppression is the increased expression of ligands for the inhibitory T cell receptors [280]. These ligands are called immune checkpoints. When binding to their inhibitory receptors on T cells, immune checkpoints suppress T cells and cause immune tolerance [280]. Immune checkpoint inhibitors (ICIs) are novel immunotherapy drugs that exhibit their effects via the blockade of the immune checkpoints and their receptors, thus restoring anti-tumor immune activity [281]. However, most patients do not respond to or develop resistance to ICIs [282,283].

Since neutrophils were recently involved in creating an immunosuppressive tumor microenvironment, they could participate in developing resistance to ICIs. TANs produce arginase, inducible nitric oxide synthase (iNOS), and CCL17 [14,28,119]. This combination may reduce the response to ICIs via the inhibition of T cells and the recruitment of Tregs into the tumor microenvironment. Based on this, targeting neutrophils along with ICIs could enhance the latter response and resolve the neutrophil-associated resistance to ICIs.

PD-1 and its ligand, PD-L1, are well-known immune checkpoint molecules that play a significant role in the suppression of anti-tumor immunity [284]. Cancer cells overexpress PD-L1 on their surface, which interacts with PD-1 on activated immune cells, especially T cells; inhibit T cell proliferation; and cause immune tolerance [124,125]. ICIs against PD-1 or its ligand are considered effective immunotherapeutic agents [285].

Besides tumor cells, PD-L1 expression was found in TANs [82,121,122,123]. In cancer patients receiving anti-PD-1/PD-L1 therapy, a high neutrophil-to-lymphocyte ratio (NLR) is thought to have prognostic and predictive value [286,287,288,289,290,291]. Furthermore, neutrophils expressing PD-L1 were linked to a poor prognosis in patients with advanced melanoma and are thought to be a novel biomarker in stage IV melanoma patients receiving anti-PD-1 immunotherapy [292]. In addition, in melanoma patients, a high NLR value was associated with anti-PD-1/PD-L1 treatment failure [293]. In a glioma animal model, neutrophil depletion improved anti-PD-1 therapy outcomes [232]. In NSCLC patients, a high level of LDNs was associated with resistance to anti-PD-1 therapy [83]. These results indicate the role of neutrophil in reducing the efficacy of the anti-PD-1/PD-L1 immunotherapy.

The combination of neutrophil-targeting agents along with ICIs was investigated in several in vivo studies. For example, CXCR2 blockade, which inhibits neutrophil recruitment to the tumor microenvironment, enhanced the efficacy of ICIs in several tumor murine models [274,276,277]. Moreover, the inhibition of NET formation with several agents (discussed below) showed significant improvements in ICI outcome [180,294,295]. Quantitative proteomic analysis of patients’ plasma revealed a possible role for the HGF-MET pathway in LDN-dependent resistance to anti-PD-1 therapy, suggesting the combination of MET inhibitors, known agents in cancer therapy, with anti-PD-1/PD-L1 therapy as an alternative approach [83,296]. A recent study found that 1-palmitoyl-2-linoleoyl-3-acetyl-rac-glycerol (PLAG) inhibited neutrophil infiltration to the tumor and normalized NLR in a mouse urothelial carcinoma model [237]. The co-administration of PLAG and anti-PD-L1 therapy has improved the latest anticancer effect, suggesting a new approach to overcome anti-PD-1/PD-L1 resistance [237]. In HCC, lactate, produced by tumor cells, was found to be involved in PD-L1 expression on neutrophils via the MCT1/NF-kB/COX-2 pathway, which inhibited the effect of lenvatinib. A COX-2 inhibitor restored lenvatinib activity, making COX-2 a potential target in cancer [238]. Kwantwi et al. showed that tumor-derived CCL20 induced PD-L1 expression in neutrophils, resulting in T cell immunosuppression, and this effect was reduced after CCL20 neutralization [123].

Faget et al. recently summarized several clinical trials in which ICIs were used in combination with drugs that could potentially affect neutrophil function and enhance ICI effects [297]. The neutrophil-targeting drugs used in clinical trials affected different sides of neutrophil biology, including neutrophil biogenesis, recruitment, and immunosuppressive functions [297].

An example of targeting neutrophil biogenesis in combination with ICIs is a second-phase clinical trial in which tocilizumab, an interleukin-6 receptor inhibitor, was investigated in combination with two checkpoint inhibitors: ipilimumab, a CTLA-4 inhibitor, and nivolumab, an anti-PD-1 monoclonal antibody, in patients with melanoma (NCT03999749). Another study targeted neutrophil recruitment using the CXCR4 antagonist BL-8040 in combination with the anti-PD-1 antibody pembrolizumab for patients with pancreatic cancer (NCT02907099). Several studies targeted neutrophil immunosuppressive potential using arginase inhibitors, COX-2 inhibitors, or iNOS inhibitors in combination with ICIs (NCT02903914, NCT03728179, and NCT03236935).

In summary, targeting neutrophils could restore the efficacy of immune checkpoint inhibitors (Figure 4).

### 4.4. Receptor Tyrosine Kinase Inhibitors

Receptor tyrosine kinase inhibitors have been used clinically in cancer therapy since 2001 [298]. Lorlatinib is an inhibitor of anaplastic lymphoma kinase (ALK) and c-ros oncogene 1 (ROS1) kinase, which is approved by the FDA for patients with ALK-positive NSCLC [299,300]. Recently, Nielsen et al. showed that neutrophils treated with conditioned medium from the pancreatic cancer cell line KPC mT4 and bone-marrow-derived neutrophils from the KPC mouse model with pancreatic tumors had increased non-receptor tyrosine kinase FES activity [233]. Since lorlatinib can also potently inhibit tyrosine kinase FES, Nielsen and colleagues investigated the potential of lorlatinib to inhibit neutrophilic FES and its contribution to tumor growth [233,301]. It was found that lorlatinib can block neutrophilic FES signaling in vitro. Moreover, lorlatinib suppresses neutrophil infiltration into the tumor and liver in the KPC mouse model, an effect combined with a reduced size of tumors and metastases. Furthermore, lorlatinib prolonged the survival of KPC mice and improved the response to anti-PD-1 immunotherapy [233]. Thus, it can be concluded that approaches to targeting TAN signaling pathways, which are enhanced during tumor initiation, are promising (Figure 4).

### 4.5. Bioactive Compounds Shifting TAN Phenotype from N2 to N1

Using synthetic or natural bioactive compounds for cancer treatment and prevention is a well-known approach [302,303]. Although different compounds could have diverse mechanisms of action, many of them were considered to modulate immune cells, including neutrophils (Table 3), against cancer [239,240,241,304,305,306]. Zhang et al. analyzed the role of neutrophil polarization in the development of tumor resistance to doxorubicin, a widely used chemotherapeutic agent. They found that doxorubicin skews HL-60 cells toward the N2 phenotype, and this shift contributes to doxorubicin resistance and promotes tumor growth [239]. Interestingly, berberine, an alkaloid from *Rhizoma coptidis* with diverse biological actions including anti-inflammatory and anti-tumor effects, was found to inhibit doxorubicin action on neutrophils and maintain the N1 phenotype, thus maintaining tumor cell sensitivity to doxorubicin [239,307]. Of note, berberine was also shown to regulate macrophage function in terms of cancer [308]. Tyagi et al. found that nicotine polarizes neutrophils to the N2 phenotype via STAT3 activation. Nicotine-polarized neutrophils can maintain breast cancer metastases into the lung and promote mesenchymal-to-epithelial transition in cancer cells primarily by secreting lipocalin-2. Tyagi et al. suggested blocking neutrophil polarization to the N2 phenotype as a candidate treatment for breast cancer lung metastasis [240]. Natural compound library screening suggested salidroside as a promising neutrophil N2 polarization inhibitor [240]. Salidroside is a glucoside of tyrosol originally isolated from the Chinese Tibetan herb *Rhodiola sachalinensis* and has diverse biological effects, including anti-tumorigenic properties [309,310].

Interestingly, salidroside inhibited neutrophil nicotine N2 polarization in vitro and significantly decreased the nicotine-mediated lung metastatic burden in a metastatic breast cancer model. Salidroside did not show toxicity in mice and did not affect cancer cell viability, which indicates its neutrophil-specific effects [240]. Li et al. investigated the effect of emodin, the main bioactive component in *Rheum palmatum*, on neutrophil function and profile in lung cancer [241]. At the beginning, HL-60 cells were differentiated into N1-like (HL-60N1) and N2-like neutrophils (HL-60N2) and were then treated with emodin. Emodin selectively induced apoptosis and decreased NET production in HL-60N2. In vivo experiments in a mouse model of urethane-induced lung cancer have shown that an increase in the number of N2 neutrophils in the alveolar cavity leads to hypercoagulation. Emodin treatment reduced hypercoagulation, which correlated with a significant decrease in N2 neutrophils in the alveolar cavity [241]. The authors also investigated the effect of emodin in the Lewis lung carcinoma model (LLC). LLC-bearing mice were treated with emodin as monotherapy or in combination with HL-60N1 or HL-60N2 cells. Emodin was able to suppress tumor growth by 20%, synergistically prevented tumor growth in combination with HL-60N1 cells, and inhibited the pro-tumorigenic actions of HL-60N2 cells [241].

Although bioactive compounds could not be a frontline therapy, the above-mentioned data shed light on different perspective compounds with neutrophil-targeted effects for investigation in cancer research (Figure 4).

### 4.6. Targeting NETs

Due to several recent studies connecting NETs to cancer initiation [172,311], progression [143,169], metastasis [56,143,179,180,312,313,314], and therapeutic resistance development [315,316], NETs have been suggested as a novel therapeutic target in cancer (Figure 4). The main strategy to target NETs is to inhibit their formation by targeting protein-arginine deiminase type 4 (PAD4). Other strategies are to inhibit different NET components (NE, MPO) or to digest NETs with DNases.

Several groups have developed PAD4 inhibitors [317,318,319,320]. Recently, Nefedova’s group has developed two new PAD4 inhibitors, BMS-P5 and JBI-589, which showed promising results in tumor mouse models [321,322]. Jiang et al. have shown that DNase I-mediated digestion of NETs formed by neutrophils, which were primed in vitro by HCC-cell-conditioned medium, led to the inhibition of the pro-migratory activity of neutrophils toward HepG2 cancer cells [313]. Neutrophils treated with HCC-cell-conditioned medium in the presence or absence of DNase I or GSK484 (a PAD4 inhibitor) were intravenously injected into mice with intrahepatic HepG2 tumors. It was found that neutrophils primed with HCC-cell-conditioned medium only (no NET inhibition) efficiently stimulated lung metastases, whereas treatment with DNase I and/or GSK484 abrogated the pro-metastatic potential of neutrophils. Moreover, in mice with intrahepatic Hepa1-6 tumors, DNase I and/or GSK484 intraperitoneal administration showed the ability to prohibit lung tumor metastasis formation [313]. In colorectal cancer models, DNase I or NE inhibitor (NEi) administration slowed tumor growth and decreased metastases to a degree comparable to that of PAD4 knockout mice [169]. In the NASH-HCC model, NET inhibition using DNase I administration or PAD4 knockout decreased tumor growth in the liver, and this effect was explained by the altering of the inflammatory environment, decreasing of Treg levels, and activity in the liver, which in turn reduced tumor burden [172,311]. In a co-culture of tumor cells with neutrophils in the presence of NET production activators (PMA/IL-8), tumor cells were covered with NETs that resulted in shielding the tumor cells from direct contact with immune effector cells and their survival in the presence of NK or cytotoxic T cells [180]. DNase I destroyed NETs and restored tumor–immune cell direct contact, leading to efficient cytotoxicity [180]. NET destruction with DNase I could also abolish NET procoagulant potential and impede cancer-associated thrombus formation [323,324,325,326].

Xia et al. designed a liver-directed gene therapy on the basis of an adeno-associated virus vector expressing DNase I (AAV-DNase I): in a metastatic colorectal cancer mouse model, AAV-DNase I injection inhibited tumor metastasis to the liver via local NET digestion [327]. Chen et al. constructed a photoregulated DNase I delivery system based on DNase I-loaded nanoparticles that were able to release the enzyme in the case of laser irradiation, and in combination therapy with anti-PD-1 therapy they showed synergistic enhanced anti-tumor effects based on tumor burden and survival rates [328]. Cheng et al. developed a DNase I-loading hydrogel with a tumor acidity neutralizer and demonstrated in an HCC mouse model that local hydrogel application after HCC resection in combination with NK infusion therapy prevented HCC recurrence [329].

Besides direct NET inhibition or digestion, many studies have attempted to identify and inactivate different molecular targets involved in NET formation or NET downstream functions. Teijeira et al. showed that supernatants from several tumor cell lines promote neutrophils to produce NETs, and this effect was abrogated after CXCR1/2 blockade [180]. Moreover, in animal models of breast and lung cancers, the production of CXCR1/2 agonists by tumor cells was associated with high NET levels in the tumors, and NET levels were decreased after CXCR1/2 blockade. These results show the potential of using CXCR1/2 inhibitors to prevent the formation of tumor-induced NETs [180].

Zhang et al. showed that IL-17 activates pancreatic cancer cells, which in turn induce pro-tumor NET formation [206]. Yang et al. demonstrated that neutrophils from HCC patients are characterized by high mitochondrial NET production (HCC-NETs). The elevated mitochondrial ROS levels in HCC neutrophils were crucial for NET production. [330]. Interestingly, oxidized mtDNA, the DNA core of HCC-NETs, triggered the expression of metastasis-promoting inflammatory mediators in HepG2 cells. Based on these findings, the authors suggested using metformin, a mitochondrial respiratory chain inhibitor, to suppress HCC-NET formation and the invasive and metastatic properties of HepG2 cells [330].

Xia et al. found that NETs support and enhance the malignant and metastatic potentials of gastric cancer cells, and these effects were reversed after cell treatment with GSK484, NEi, or DNase I [179]. Tumor-supporting effects of NETs were realized via the activation of the TGF-β signaling pathway in cancer cells. Based on this, in an animal model of metastatic gastric cancer, the pro-tumorigenic effects of NETs were abolished upon treatment with DNase I or galunisertib, an inhibitor of the TGF-β pathway [179].

Several investigators have attempted to figure out how NET inhibition could interact with other already established therapeutic approaches in pre-clinical studies. In tumor animal models, NET inhibition in combination with anti-PD-1 therapy was shown to improve the efficacy of therapeutic approaches [180,206,294,295]. In the MC38-induced colorectal cancer model, anti-PD-1 or DNase I monotherapies were able to retard tumor growth. At the same time, the combination therapy showed significant improvements in survival rate and tumor volume reduction. This effect was realized most likely via the digestion of NETs with DNase I which resulted in the reversal of anti-PD-1 blockade resistance through enhancing CD8+ T cell infiltration and cytotoxicity [294].

In the 4T1-induced breast cancer model, the combination therapy with GSK484 and dual checkpoint blockade of PD-1 and cytotoxic T-lymphocyte-associated protein 4 (CTLA-4) showed synergistic enhancement of the effects of both monotherapies [180]. In a pancreatic cancer mouse model, IL-17 blockade, neutrophil depletion, and PAD4 knockout were able to enhance anti-PD-1 treatment [206]. Another study found that using exenatide in combination with anti-PD-1 therapy improved outcomes, with the effects realized through the inhibition of NET production [295].

NETs were shown to be involved in bladder cancer resistance to radiation therapy [316]. In the invasive MB49-induced urothelial carcinoma mouse model, radiation therapy was shown to promote neutrophil infiltration and NET production in the tumor. NET inhibition using PAD4 knockout, NEi gavage, or intramuscular injection of DNase I, in combination with radiation therapy, delayed tumor growth and improved mouse survival [316]. Further experiments showed that radiation therapy promotes NET production via cancer-derived HMGB1 interacting with TLR4 on neutrophils, and the inhibition of HMGB1 or NETs showed good results in resolving radiation therapy resistance [316]. Of note, neutrophilic HMGB1 integrated with NETs is involved in EMT induction in cancer cells [312]. Thus, inhibiting HMGB1 stops its loop of action, affecting both cancer cells and neutrophils.

Overall, inhibiting NETs may reduce their pro-tumor actions and improve the outcomes of other cancer therapies. However, the pro-tumorigenic role of NETs does not conceal their vital role in terms of infections, which forces researchers to further investigate NET inhibitors and their effects upon infectious conditions [331].

### 4.7. Anti-Tumor NEVs

Recently, much has been said about the prospects of using EVs for therapy. EVs from immune cells may be a particularly interesting strategy [192]. As stated above, NEVs have interesting properties; in particular, they can carry stimulatory factors and are potentially capable of killing pathogens [215]. Moreover, NEVs have a short life span and are easy to handle, making them very advantageous for use as drug carriers [192].

The therapeutic potential of NEVs was recently demonstrated in a mouse model of rheumatoid arthritis [332]. In this study, the delivery of annexin A1^+^ NEVs to the knee joint prevented cartilage damage through the FPR2-dependent generation of TGF-β [332]. Additionally, Wang et al. recently reported that drug-carrying neutrophil EVs can rapidly cross the blood–brain barrier and migrate to the brain [333]. Intravenous injection of doxorubicin-loaded NEVs effectively suppressed tumor growth and prolonged the survival in a mouse model of glioma [333]. In addition, in some cases, neutrophils can produce NEVs that induce macrophage polarization toward a pro-inflammatory phenotype [334]. The most impressive results were obtained by Xu Zhang’s group, which developed specially engineered NEVs that have a cytotoxic effect on tumor cells by activating the apoptosis signaling pathway [217]. In addition to studies showing the therapeutic potential of native NEVs as a standalone therapeutic agent, to achieve a higher tumor-targeting therapeutic effect, NEVs loaded with doxorubicin and decorated with superparamagnetic iron oxide nanoparticles (SPIONs) were investigated [217]. The authors demonstrated that these NEVs exhibited a dual therapeutic effect achieved through the delivery of a cytostatic agent to tumor cells and the immune functions of the NEVs that almost abolished tumor growth in mice [217].

### 4.8. CAR-Neutrophils

T cells are so far considered the pioneers in the CAR therapy area of research. However, CAR-T cell therapy has not yet been applied to solid tumors and suffers from different problems, such as CAR-T cell immunosuppression in the tumor microenvironment and high toxicity [335]. To overcome these difficulties, myeloid cells could be an alternative to T cells or a supportive factor in CAR therapy. Roberts et al. published the first report on the use of neutrophils in CAR therapy in 1998 [336]. The authors developed neutrophils with anti-HIV-specific CD4ζ chimeric receptors from hemopoietic stem cells. The transduced neutrophils showed improved cytotoxicity against tumor cells transfected with the HIV envelope [336]. Recently, Chang et al. developed neutrophils with glioblastoma-targeting CAR from human pluripotent stem cells, which displayed enhanced anti-tumor cytotoxicity both in vitro and in vivo [17]. Despite the challenges in this field, CAR-neutrophils represent a novel option in CAR-based cancer therapy (Figure 4).

## 5. Conclusions

In comparison with other leukocyte types, the role of neutrophils in cancer is a newly established research area with many questions that are still waiting to be answered. The slow progress in this field is most likely due to several technical challenges when investigating neutrophils, particularly their spontaneous activation and short life span in vitro. According to one perspective, the newly described neutrophil heterogeneity and plasticity could be viewed as a new challenge in studying the neutrophil role in cancer. From another angle, neutrophil heterogeneity is the property that makes neutrophils a promising target in cancer therapy. This sheds light on new perspectives that can be implemented in the fight against cancer with the help of the most abundant leukocyte in human blood, the neutrophil. Based on the information available so far, potential therapeutic options include inhibiting neutrophil polarization to a pro-tumor phenotype (N2) or weakening N2-polarized neutrophil effects (TGF-inhibitors). Inhibiting NETosis or digesting NETs with DNase could also be implemented. Approaches to reduce the immunosuppressive effect of neutrophils are also being established (COX and iNOS inhibitors). Another strategy is to inhibit neutrophil recruitment to the tumor microenvironment (CXCR2 antagonists). However, this strategy is only useful if a pro-tumor neutrophil phenotype is detected. Moreover, in tumors with high infiltration of neutrophils, drug delivery by neutrophils could be a viable approach. However, the most promising option is to reprogram neutrophils toward an anti-tumor phenotype and enhance their anti-tumor activities. Attempts have already been made to shift neutrophils to the N1 phenotype using interferons and bioactive compounds. In addition, the idea of CAR-neutrophil generation in an attempt to obtain cytotoxic and targeted neutrophils is already on the table. Nevertheless, future studies are needed to solve the puzzle and capture the whole picture of the complicated tumor–neutrophil connections in order to suggest novel neutrophil-based cancer therapies.

## Figures and Tables

**Figure 4 ijms-23-15827-f004:**
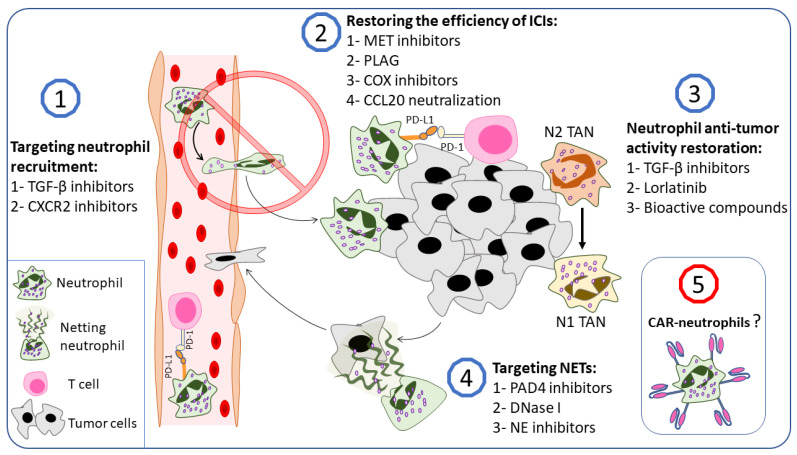
Neutrophil in cancer therapy: potential approaches. Several neutrophil-based anticancer therapies have recently been investigated: (**1**) A suggested strategy is to inhibit neutrophil recruitment to the tumor microenvironment. TGF-β and CXCR2 inhibition is the first strategy that comes to mind, since they are widely involved in neutrophil recruitment to the tumor microenvironment [26,236]. More promising strategies are to block neutrophil immunosuppressive function or to restore neutrophil anti-tumor properties. (**2**) MET inhibitors [83], PLAG [237], COX inhibitors [238], and CCL20 inhibitors [123] could inhibit neutrophil immunosuppressive functions and restore the efficiency of ICIs. (**3**) To restore neutrophil anti-tumor activities, TGF-β inhibitors [14], lorlatinib [233], and some selected bioactive compounds (berberine [239], salidroside [240], and emodin [241]) are considered reliable choices. (**4**) NET inhibition is also a potential therapeutic approach that could be applied by the inhibition of NET production (PAD4 inhibitors [235]), the digestion of NETs (DNase I [235]), or the inhibition of different NET compounds (NE inhibitors [169]). (**5**) Recently, CAR-neutrophils were developed as a novel approach to use neutrophils in cancer therapy [17]. TGF-β—transforming growth factor beta, CXCR—CXC chemokine receptor, PD-L1—programmed death-ligand 1, PD-1—programmed cell death protein 1, MET—mesenchymal–epithelial transition tyrosine kinase receptor, PLAG—1-palmitoyl-2-linoleoyl-3-acetyl-rac-glycerol, COX—cyclooxygenase, CCL—C-C motif chemokine ligand, ICIs—immune checkpoint inhibitors, NET—neutrophil extracellular trap, PAD4—protein-arginine deiminase type-4, NE—neutrophil elastase, CAR—chimeric antigen receptor.

**Table 3 ijms-23-15827-t003:** Natural compounds with demonstrated effects on neutrophil polarization.

Natural Compound	Formula	Natural Source	Observed Effects on Neutrophils	Possible Mechanism of Action	Reference
Berberine	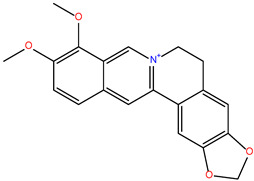	*Rhizoma coptidis*	Inhibits doxorubicin-induced neutrophil N2 polarization and maintains N1 phenotype	Regulation of JAK-STAT and FoxO signaling pathways	[239]
Salidroside	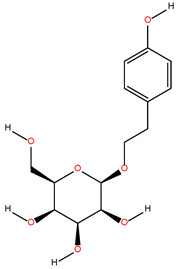	*Rhodiola sachalinensis*	Inhibition of N2 polarization induced by nicotine	STAT3 inhibition	[240]
Emodin	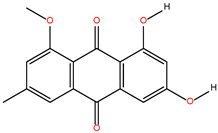	*Rheum palmatum*	Selectively suppresses N2 neutrophils	Regulation of IL-10, TLR4, START3, and CCL2 expression	[241]

## Data Availability

Not applicable.

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
