# Peer review of "Diverse Neutrophil Functions in Cancer and Promising Neutrophil-Based Cancer Therapies"

_ijms, 2022, doi:10.3390/ijms232415827_

Round 1

Reviewer 1 Report

The review article extensively covers almost all the aspects and the recent developments about the role of neutrophils in the the cancer complexities and possible therapeutic strategies. 

The in depth discussion of the changes at the molecular level provides mechanistic insights and distinguishes this article from other general reviews  in the area.

Table 1 is highly informative and summarizes all the relevant information.

The structure of the review article is well organized. The illustrations are helpful but may need more descriptive figure legends

However, it may further increase the readability if some of the extensive  text could  be edited / removed to give it a crispier look. At present there appears to be lack of focus.

There are several English language errors and long indirect sentences and the titles for ex "Neutrophil-based perspective anticancer strategies" which are compromising the clarity and not conveying the real meaning. It is suggested to get the full text checked and edited by an English language professional. 

Author Response

To Reviewer 1

Dear reviewer 1, we thank you for your comments that helped us to improve our manuscript. We revised the manuscript according to your comments. Revised parts are marked up using the “Track Changes” function.

1- “The review article extensively covers almost all the aspects and the recent developments about the role of neutrophils in the the cancer complexities and possible therapeutic strategies. 

The in depth discussion of the changes at the molecular level provides mechanistic insights and distinguishes this article from other general reviews  in the area.

Table 1 is highly informative and summarizes all the relevant information.”

Answer: We appreciate your positive comments on our manuscript.

2- “The structure of the review article is well organized. The illustrations are helpful but may need more descriptive figure legends”

Answer: We edited the figure legends, especially for figure 2 as some information was missing. Moreover, we added a list of abbreviations for each figure.

3- “However, it may further increase the readability if some of the extensive  text could  be edited / removed to give it a crispier look. At present there appears to be lack of focus.”

Answer: The manuscript was edited and extensive text in different paragraphs was removed. We also noticed the lack of focus in section 2 (Neutrophil heterogeneity in cancer: N1/N2, NDN/LDN, and g-MDSC). We divided this section into three separate sections (2.1 N1 vs N2, 2.2. NDN vs LDN and 2.3. g-MDSCs) and removed the information that is not related to these classifications.

3- “There are several English language errors and long indirect sentences and the titles for ex "Neutrophil-based perspective anticancer strategies" which are compromising the clarity and not conveying the real meaning. It is suggested to get the full text checked and edited by an English language professional.”

Answer: To improve the English and readability, our manuscript has been edited using English editing and proofreading web-service TRINKA (https://www.trinka.ai/). The title “Neutrophil-based perspective anticancer strategies” was changed to “Neutrophil in cancer therapy: perspective strategies”. Other titles were also modified to a simpler variant for example “Exchange of extracellular vesicles between tumor cells and neutrophils” was changed to “Tumor cells and neutrophils exchange extracellular vesicles”.

Reviewer 2 Report

General comment

The manuscript entitled “Neutrophil diverse functions in cancer and promising neutrophil-based cancer therapies” aims to summarize the current view on neutrophil heterogeneity in cancer, reviewing in addition the different communications pathways between tumor and neutrophils. The manuscript is quite well written and very comprehensive. This is double edged sword though as in some sections the paragraphs seems too long and somewhat general. In order to improve the quality of your work, I would like to suggest few corrections:

-          Regarding the style of writing, references should be placed at the end of the sentences instead mid-sentence.

-          The second paragraph is way too long. Try to collect and cull the excessive data or divide the paragraph in smaller ones.

-          Check English grammar along the text as few typos and incorrect grammar is present.

-          For the third paragraph, a table or a figure summarizing the reported data could be a nice addition.

-          Regarding the forth paragraph and the role of neutrophils in anti-cancer therapies, it would be interesting to focus on the role of new therapies and neutrophils in response to immunotherapy. Please see: DOI: 10.3390/cancers14102545 and DOI: 10.1136/jitc-2020-002242

Author Response

To Reviewer 2

Dear reviewer 2, we appreciate your positive comments about the manuscript and thank you for your valuable suggestions that helped us to improve our manuscript. We revised the manuscript according to your comments. Revised parts are marked up using the “Track Changes” function.

1- “Regarding the style of writing, references should be placed at the end of the sentences instead mid-sentence.”

Answer: We agree that mid-sentence references seemed to be confusing in some cases. Wherever it was confusing, the references were replaced to the end of the sentences.

2-The second paragraph is way too long. Try to collect and cull the excessive data or divide the paragraph in smaller ones.”

Answer: We divided this paragraph into three separate sections (2.1 N1 vs N2, 2.2. NDN vs LDN and 2.3. g-MDSCs) and removed the information that is not related to these classifications.

3- “Check English grammar along the text as few typos and incorrect grammar is present.”

Answer: Checked. The manuscript has been edited using English editing and proofreading web-service TRINKA (https://www.trinka.ai/).

4- “For the third paragraph, a table or a figure summarizing the reported data could be a nice addition.”

Answer: In the third paragraph the actions of neutrophils toward tumor cell were summarized in Figure 2. (Mechanisms of anti-tumor (A) and pro-tumorigenic (B) activities of neutrophils). Since how tumor cells affect neutrophils was lacking, we added a figure 3 summarizing the data mentioned in sections 3.3, 3.4 and 3.5.

5- “Regarding the fourth paragraph and the role of neutrophils in anti-cancer therapies, it would be interesting to focus on the role of new therapies and neutrophils in response to immunotherapy. Please see: DOI: 10.3390/cancers14102545 and DOI: 10.1136/jitc-2020-002242”

Answer: Thank you for the helpful references. We agree that the newly described immunosuppressive activities of neutrophils raised the interest to investigate the role of neutrophils and target them in combination with immunotherapy, especially with immune checkpoint inhibitors. To address this issue more detailed, the following changes were done:

  • the title of the section “4.3 PD-L1 blockade” was modified to “Targeting neutrophils to restore the efficiency of immune checkpoint inhibitors”.
  • the question “why neutrophils should be studied in the field of immunotherapy?” was discussed more fully.
  • Examples of the in vivo studies and the recent clinical trials in this field were added.
